# EARL-BO: Reinforcement Learning for Multi-Step Lookahead, High-Dimensional Bayesian Optimization

**Mujin Cheon** [1 2]  **Jay H. Lee** [3]  **Dong-Yeun Koh** [2]  **Calvin Tsay** [1]

## Abstract

To avoid myopic behavior, multi-step lookahead Bayesian optimization (BO) algorithms consider the sequential nature of BO and have demonstrated promising results in recent years. However, owing to the curse of dimensionality, most of these methods make significant approximations or suffer scalability issues. This paper presents a novel reinforcement learning (RL)-based framework for multi-step lookahead BO in high-dimensional black-box optimization problems. The proposed method enhances the scalability and decision-making quality of multi-step lookahead BO by efficiently solving the sequential dynamic program of the BO process in a near-optimal manner using RL. We first introduce an Attention-DeepSets encoder to represent the state of knowledge to the RL agent and subsequently propose a multi-task, fine-tuning procedure based on end-to-end (encoder-RL) on-policy learning. We evaluate the proposed method, EARL-BO (Encoder Augmented RL for BO), on synthetic benchmark functions and hyperparameter tuning problems, finding significantly improved performance compared to existing multi-step lookahead and high-dimensional BO methods.

## 1. Introduction

Optimization of an unobserved, "black-box" function underlies engineering applications such as hyperparameter tuning (Snoek et al., 2012), material discovery (Shahriari et al., 2015), reaction chemistry (Folch et al., 2022), energy systems (Thebelt et al., 2022), and robotics (Muratore et al.,

2021). Many of these settings involve costly data acquisition (i.e., experiments) and no information about the gradient. Bayesian Optimization (BO) is popular in such problems due to its ability to query data-efficient samples (Brochu et al., 2010; Paulson & Tsay, 2024). BO leverages a surrogate model with uncertainty quantification, typically a Gaussian process, or GP (Williams & Rasmussen, 2006), in conjunction with an acquisition function, which contains the "philosophy" about how to balance exploration and exploitation, to identify and optimize the underlying function in a sequential sampling process.

Traditional BO algorithms deploy acquisition functions such as Expected Improvement (Jones et al., 1998), Probability of Improvement (Kushner, 1964), and Upper Confidence Bound (Srinivas et al., 2010) in one-step lookahead policies. In other words, these policies ignore the effect(s) of the choice made at current iteration on future steps and instead solely focus on the immediate maximization of the acquisition function. Previous studies (Ao & Li, 2024; Lam et al., 2016; Osborne et al., 2009; Wu & Frazier, 2019) have demonstrated that these so-called *myopic* policies can result in sub-optimal performance. Moreover, anticipating the sequential nature of BO can enable consideration of the path of samples taken (Folch et al., 2022; Yang et al., 2024).

Along these lines, Ginsbourger & Le Riche (2010) observe that the decision-making process of BO can be seen as a partially observable Markov decision process (POMDP), and therefore policies can be improved by multi-step lookahead decisions. Several lookahead methods have since been proposed. GLASSES (González et al., 2016) involves approximating the ideal lookahead loss function in an open-loop approach. Rollout-based strategies (Lam et al., 2016; Lee et al., 2020; Osborne et al., 2009) approximate the POMDP value function by "rolling-out" heuristic decision-making policies (e.g., maximizing EI) over future time steps to approximate long-term gains. In this category, Wu & Frazier (2019) suggest practical two-step lookahead methods using an envelope-function estimator to improve computational tractability. More recently, Cheon et al. (2024) introduce a reinforcement learning (RL) strategy to solve the multi-step lookahead POMDP of BO in a near-optimal way.

Multi-step lookahead methods generally face two critical

[1]Department of Computing, Imperial College London, UK [2]Department of Chemical and Biomolecular Engineering, Korea Advanced Institute of Science & Technology (KAIST), South Korea [3]Mork Family Department of Chemical Engineering and Materials Science, University of Southern California, USA. Correspondence to: Calvin Tsay <c.tsay@imperial.ac.uk>.

*Proceedings of the 42$^{nd}$ International Conference on Machine Learning*, Vancouver, Canada. PMLR 267, 2025. Copyright 2025 by the author(s).

challenges: they must (1) simplify the Stochastic Dynamic Programming (SDP) problem, leading to a sub-optimal policy, and/or (2) suffer from scalability issues that hinder applicability to larger and higher-dimensional problems.

Given these challenges, this work introduces a novel framework combining end-to-end encoder architectures with reinforcement learning to create a scalable, RL-based BO framework tailored for multi-step lookahead optimization in higher-dimensional, black-box optimization problems. Our Encoder Augmented RL for BO (EARL-BO) framework leverages the representational power of an Attention-DeepSets-based encoder to parameterize the current knowledge of the state within a latent space that is amenable to RL. The RL agent learns in a model-based fashion, using a GP virtual environment to receive rewards based on multi-step performance, completing the end-to-end learning structure. The proposed integration enables the RL algorithm to solve the SDP of the BO decision-making process in a near-optimal, yet tractable way over an extended horizon.

In summary, our contributions include:

- An integrated and modular RL-based BO method that efficiently solves the SDP inherent in multi-step BO.

- The introduction of an Attention-DeepSets based encoder, which provides a scalable representation of the knowledge state in BO, enhancing the model's ability to handle high-dimensional spaces.

- An end-to-end, model-based learning procedure for the combined encoder-RL pair using a GP-based virtual environment.

- Comprehensive evaluations of EARL-BO across both synthetic benchmark functions and real-world hyper-parameter tuning, against other multi-step lookahead and high-dimensional optimization methods.

Our work advances the state of the art in non-myopic BO by addressing the limitations of existing approaches and providing a scalable and modular framework for high-dimensional black-box optimization problems.

## 2. Background and Related Works

Consider the problem of globally maximizing a continuous function $f(x)$ defined over a compact domain $\Omega \subset \mathbb{R}^d$. When evaluating $f(x)$ is costly, it becomes crucial to minimize the number of function evaluations. BO uses a Gaussian process (GP) to model the objective function $f(x)$. Based on the GP prior $f(x) \sim GP(\mu, K)$, where $\mu : \Omega \to \mathbb{R}$ is the mean function and $K : \Omega \times \Omega \to \mathbb{R}$ is the covariance (or kernel) function that encodes the correlation

between nearby points (e.g., assumptions about the smoothness of the function), and a dataset $\mathcal{D}_k = \{(x_i, y_i)\}_{(i=1)}^k$, the posterior distribution of the function value at a new query point $x$ is Gaussian, $\mathcal{N}(\mu^k(x; \mathcal{D}_k), K^k(x, x; \mathcal{D}_k))$, where:

$$\mu^k(x; \mathcal{D}_k) = \mu(x) + \mathbf{k}(x)^T (K + \sigma^2 I_k)^{-1} (y - \mu(x))$$
$$K^k(x, x; \mathcal{D}_k)) = K(x, x) - \mathbf{k}(x)^T (K + \sigma^2 I_k)^{-1} \mathbf{k}(x).$$

Here, $I_k$ denotes the $k \times k$ identity matrix. The mean $\mu^k(x; \mathcal{D}_k)$ represents the posterior prediction of the function at $x$, and $K^k(x, x; \mathcal{D}_k))$ reflects the covariance.

The subsequent point $x_{k+1}$ for evaluation is selected by optimizing an acquisition function $\Lambda(x|\mathcal{D}_k)$, such that:

$$x_{k+1} = \underset{x \in \Omega}{\operatorname{argmax}} \; \Lambda(x|\mathcal{D}_k). \tag{1}$$

For example, one popular acquisition function, Expected Improvement (EI), is defined as:

$$\Lambda_{EI}(x) = \mathbb{E}\left[\max(0, f(x) - f(x^+))\right], \tag{2}$$

where $f(x^+)$ is the current best observation. The point $x$ that maximizes the acquisition function $\Lambda(x)$ is selected as the next evaluation point, noting that this optimization problem itself may be non-trivial to solve (Ament et al., 2023; Xie et al., 2024). While this *one-step* lookahead strategy can be effective in many scenarios, it is inherently myopic, focusing only on the immediate benefits and neglecting the long-term effects of decisions.

### 2.1. Multi-step Lookahead Approaches

To overcome the limitations of one-step lookahead methods, multi-step lookahead strategies seek to account for the impacts of current decisions on subsequent evaluations.

**Rollout-Based Bayesian Optimization.** Rollout methods are a class of multi-step lookahead strategy, where future decisions are approximated using a base policy $\pi_b$. This base policy could involve applying a heuristic or a computationally simpler, sub-optimal strategy, such as a policy that maximizes EI (2). The objective is to enhance the base policy by maximizing the cumulative expected reward over a user-defined horizon $H$:

$$\Lambda_{\text{rollout}}(x|\mathcal{D}_k) = \mathbb{E}^{\pi}_{y_{k+1}, \dots, y_{k+H}}\left[\sum_{n=1}^{H} r(x_{k+n}, y_{k+n})\right].$$

Here, $r(x_{k+n}, y_{k+n})$ is the reward at step $(k + n)$, which is often defined as the improvement in the objective function, and $\mathbb{E}^{\pi}[\cdot]$ denotes the expectation over possible future trajectories given a policy $x_{k+1} \sim \pi(\mathcal{D}_k)$, with $y_{k+1} = f(x_{k+1})$.

The seminal work of Osborne et al. (2009) demonstrates the possibility of computing this rollout acquisition function

for a 1-D optimization problem with a two-step lookahead horizon (H=2). Later, Lam et al. (2016) reduce the computational burden by approximating the expected future rewards using Gauss-Hermite quadrature, enabling good performance on 2-D synthetic functions with up to five-step lookahead horizons. More recently, Lee et al. (2020) suggest rollout-based BO with variance reduction, i.e., reducing the uncertainty (variance) in the predictions made during each rollout to produce more reliable estimates of future rewards. The authors demonstrate this strategy on traditional BO methods such as EI, UCB, KG in 2-D and 4-D benchmark functions with up to six-step lookahead horizons.

Despite the benefits of rollout-based BO and the above strategies, several challenges remain:

- **Computational Complexity:** Even with variance reduction, the computational cost of evaluating multi-step lookahead strategies is substantial, particularly as the dimensionality of the search space increases.

- **Approximation Errors:** The need to approximate future decisions using a base (heuristic) policy will inevitably lead rollout methods to sub-optimal policies.

To mitigate the latter approximation errors, Cheon et al. (2024) use RL to make multi-step lookahead decisions, showing improved performance compared to rollout-based methods on 2-D benchmark functions. However, the algorithm employs a latticized representation of a GP as the RL agent input, which suffers from the curse of dimensionality.

Noting that the above works are limited to low-dimensional (approx. ≤4-D) problems, this work focuses on multi-step lookahead BO for higher-dimensional search spaces. Due to the computational infeasibility of rollout methods for such settings, we compare performance against rollout-based BO for low-dimensional problems but against scalable (one-step lookahead) BO methods for higher-dimensional problems.

### 2.2. High Dimensional Approaches

Several recent works address challenges posed by increasing dimensionality in BO (Wang et al., 2023). Hoang et al. (2018) propose the Decentralized High-dimensional Bayesian Optimization (DEC-HBO) algorithm, which employs a sparse factor graph representation of the objective function to exploit interdependent effects of input components while maintaining scalability. The authors propose a decentralized strategy for optimizing the acquisition function and demonstrate DEC-HBO's performance on various tasks, including synthetic functions of up to ten dimensions.

Another popular method is Trust Region Bayesian Optimization, or TuRBO (Eriksson et al., 2019). TuRBO partitions the search space into multiple local regions, each defined as a trust region and modeled with GP. This approach has been extensively evaluated on diverse high-dimensional synthetic and real-world problems, such as the 200-dimensional Ackley function and robot control tasks, where TuRBO outperforms traditional BO methods and alternative optimization techniques, including evolutionary algorithms.

SAASBO (Eriksson & Jankowiak, 2021) introduces Sparse Axis-Aligned Subspaces to effectively handle the curse of dimensionality. By learning which dimensions are most relevant to the optimization objective, SAASBO can focus on the most influential parameters and scale to problems with hundreds of dimensions. Given their relative popularity and strong performance, we use TuRBO and SAASBO as benchmarks for comparison in higher-dimensional problems.

In a similar vein, VAE-BO methods address high-dimensional BO using Variational Autoencoders (VAEs) to learn low-dimensional latent representations of the search space (Gómez-Bombarelli et al., 2018). LBO (Tripp et al., 2020) improves upon this by incorporating weighted retraining of the VAE, assigning higher weights to better-performing points and periodically updating the VAE. More recently, Chen et al. (2024) propose PG-LBO, which introduces pseudo-label training to leverage unlabeled data and GP guidance to directly integrate labeled data.

### 2.3. Learning to Optimize (L2O)

This work falls within the broader framework of Learning to Optimize (L2O), an emerging paradigm that leverages machine learning techniques to improve traditional optimization methods. Specifically, instead of relying on predefined heuristics, L2O approaches learn optimization strategies from data, enabling them to adapt to different problem classes and potentially discover superior optimization behaviors (Chen et al., 2017; Ma et al., 2025).

Recent advances in L2O for black-box optimization include RIBBO (Song et al., 2024), which employs transformer architectures with regret-to-go tokens to learn end-to-end algorithms from offline datasets, and POM (Li et al., 2024), which combines population-based optimization with transformer encoders for zero-shot black-box optimization. A more comprehensive review of L2O can be found in Ma et al. (2025). While these methods demonstrate promising results, they again generally remain fundamentally limited to single-step, myopic decision making (the above approaches generate query points based solely on the current state).

## 3. Methodology

### 3.1. Preliminaries

**Bayesian Optimization as MDP.** Lam et al. (2016) conceptualize the decision-making process of BO as a finite-

horizon dynamic program. The key idea is that, given a set of observed data $\mathcal{D}_k$, the data-acquisition order and procedure are irrelevant as long as the same data points are obtained. Thus, the BO setting satisfies the Markov property. Here, we express this framework using the equivalent Markov Decision Process (MDP) formulation, adopting the standard notation from Puterman (2014).

An MDP is defined by the tuple $\langle T, S, A, P, R \rangle$:

- $T$ is the set of decision epochs, $T = \{0, 1, \ldots, h - 1\}$, with $h < \infty$ here representing a finite horizon.

- $S$ is the *state space* comprising the information necessary to describe the system at any given time $t \in T$.

- $A$ is the *action space* of possible decisions.

- $P(s'|s, a)$ is the *transition probability*, or the likelihood of moving to $s'$ by taking action $a$ at state $s$.

- $R(s, a, s')$ defines the *reward* received when transitioning from state $s$ to state $s'$ by taking action $a$.

Following this notation, a decision rule $\pi_t : S \to A$ maps states to (a distribution of) actions at time step $t$. A policy $\pi = [\pi_0, \pi_1, ..., \pi_{h-1}]$ is a sequence of decision rules, one for each decision epoch. Given a policy $\pi$, an initial state $s_0$, and a horizon $h$, the expected total reward $V_h^\pi(s_0)$ is:

$$V_h^\pi(s_0) = \mathbb{E}\left[\sum_{t=0}^{h-1} R\left(s_t, \pi_t(s_t), s_{t+1}\right)\right]. \qquad (3)$$

In this MDP framework, the objective is to determine the optimal policy $\pi^*$ that maximizes the expected total reward:

$$\pi^* = \underset{\pi \in \Pi}{\operatorname{argmax}} \, V_h^\pi(s_0), \qquad (4)$$

where $\Pi$ is the set of all possible policies. This formulation allows us to view BO as a sequential decision-making problem, where the goal is to maximize the cumulative reward over a finite horizon by selecting the best possible actions at each step. Moreover, Paulson & Tsay (2024) observe that choosing an acquisition function $\Lambda$ can be mapped to a reward function from the viewpoint of dynamic programming.

For the BO setting, $\langle T, S, A, P, R \rangle$ can be construed as $T$: a user-intended lookahead horizon for BO, $S$: data acquired in the BO process ($\mathcal{D}_t$), $A$: the next query point of BO, $P$: the change in state when a point $x_{t+1}$ is queried, and $R$: a user-defined reward, which, analogously to (2), is commonly chosen as $R(\mathcal{D}_t, x_{t+1}, \mathcal{D}_{t+1}) = \max(0, y_{t+1} - y_t^*)$ for multi-step lookahead BO, where $y_t^*$ denotes the maximum value of $y$. Notice that directly using the obtained data $\mathcal{D}_t$ as the state requires continually increasing the dimensionality of the state vector as BO proceeds. We therefore

seek to represent data in a size- and permutation-invariant way using a tailored encoder structure.

**Reinforcement Learning (RL).** RL is a paradigm in machine learning where an agent learns to make decisions by interacting with an environment (Sutton & Barto, 2018). In our context, we employ RL to learn an optimal policy for selecting query points, i.e., solving the MDP formulation of the BO problem. Several studies use RL to learn such a policy, e.g., by meta-learning acquisition functions for GPs (Hsieh et al., 2021; Volpp et al., 2019). Later, Shmakov et al. (2023) integrate learning of deep kernels, and Maraval et al. (2024) propose an end-to-end framework based on meta-learning both a transformer and an RL policy. These works are similar in spirit, but consider the L2O setting of meta-learning across problems (Chen et al., 2017). On the other hand, EARL-BO aims to learn to emulate multi-step lookahead BO on the given task.

In this work, we focus on the general setting where data from other tasks are not available for meta-learning, and instead take a model-based RL approach. Our framework is modular and agnostic to the specific choice of RL algorithm; here we employ Proximal Policy Optimization (PPO), a policy gradient algorithm that addresses the challenges of step-size selection and sample efficiency (Schulman et al., 2017). PPO uses a clipped surrogate objective to ensure that policy updates are not overly large (which could otherwise lead to performance collapse), defined as:

$$\begin{aligned} L^{\text{PPO}}(\theta) = \mathbb{E}_t\Big[\min\big(r_t(\theta)A_t, \\ \operatorname{clip}(r_t(\theta), 1 - \epsilon, 1 + \epsilon)A_t\big)\Big], \end{aligned} \qquad (5)$$

where:

- $r_t(\theta) = \frac{\pi_\theta(a_t|s_t)}{\pi_{\theta,old}(a_t|s_t)}$ is the probability ratio between the new and old policies,

- $A_t$ is the estimated advantage at time $t$, and

- $\epsilon$ is a hyperparameter that controls the clip range.

The clipping in the objective function (often implemented with $\epsilon = 0.2$) ensures that the employed ratio between the new and old policies does not deviate significantly from unity, which helps to stabilize training in practice.

### 3.2. Overview of EARL-BO

EARL-BO is a modular framework comprising: (1) an encoder module that learns a representation of the state of data acquisition and (2) an RL module that learns the lookahead BO policy. An overview of EARL-BO is shown in Figure 1. The framework is agnostic to the specific choice of encoder and RL algorithm. Similarly, the off-policy learning step described below can leverage any existing BO method.

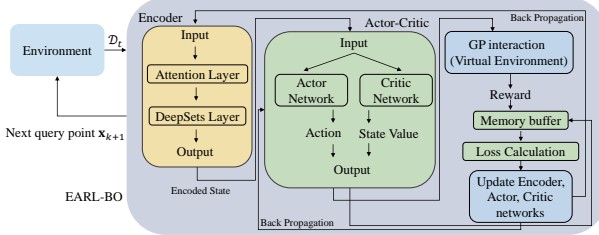

*Figure 1.* An overview of the EARL-BO architecture.

**Attention-Deepsets Encoder Module.** Permutation invariance is crucial in BO, as the order in which data points is acquired should not affect the learning process (i.e., the MDP). In other words, given a dataset $\mathcal{D}_k = \{(x_i, y_i)\}_{i=1}^k$, the order of the data pairs $(x_i, y_i)$ should not influence the agent's decision-making process. While meta-learning studies have leveraged transformer models (Chen et al., 2022; Maraval et al., 2024), we begin with the simpler DeepSets architecture (Zaheer et al., 2017), which ensures that the encoding function $\phi$ applied to the data is invariant under permutations. Mathematically, this is expressed as:

$$\phi(\mathcal{D}_t) = \rho\left(\sum_{(x_i, y_i) \in \mathcal{D}_t} \psi(x_i, y_i)\right), \qquad (6)$$

where $\psi$ is a function that maps individual data points to a latent space, and $\rho$ is a function that aggregates these representations into a fixed-size vector. This aggregation achieves permutation invariance of the final representation.

Another requirement for the encoder is size invariance, which refers to the ability to handle a growing dataset $\mathcal{D}_t$ as samples are acquired. As new data points are added, the encoder must adapt without being explicitly retrained or altered. The DeepSets architecture inherently provides this property by summing over the data representations, allowing it to naturally handle datasets of varying sizes. Note that the aggregation in (6) could be replaced with other permutation-invariant functions, e.g., arithmetic mean.

While permutation and size invariance are important, data points may not contribute equally to the decision-making process in BO: some points may be more relevant than others in determining the next query point. We therefore incorporate an attention mechanism (Vaswani et al., 2017), which assigns weights to each data point based on its learned relevance, effectively allowing the model to "focus" on the critical information (Gordon et al., 2020; Simpson et al., 2021).

Mathematically, the attention weights $\alpha_i$ for each data point

$(x_i, y_i)$ are computed as:

$$\alpha_i = \text{softmax}\left(f_{\text{att}}(\psi(x_i, y_i))\right), \qquad (7)$$

where $f_{\text{att}}$ is a scoring function that measures the relevance of each data point. The final output is then computed as:

$$\phi_{\text{att}}(\mathcal{D}_t) = \rho\left(\sum_{(x_i, y_i) \in \mathcal{D}_t} \alpha_i \cdot \psi(x_i, y_i)\right). \qquad (8)$$

By using an Attention-DeepSets based encoder, we ensure that the state representation used in EARL-BO is permutation invariant, size invariant, and capable of focusing on the salient information in the dataset. This robust encoding strategy is essential for effectively navigating the high-dimensional search space in multi-step lookahead BO tasks.

**PPO Actor-Critic Module.** The core of EARL-BO's decision-making process is the RL module. We employ PPO, which comprises two main components: the actor network and the critic network.

The *actor* network, denoted as $\pi_\theta(a|s)$ with parameters $\theta$, represents the policy that maps states to actions. In our case, it outputs the next query point for the BO process. The actor's objective is to maximize the expected cumulative reward, (3). The critic network, denoted as $V_\phi(s)$ with parameters $\phi$, estimates the value function of the current state and helps reduce the variance of policy gradient estimates by providing a baseline for advantage estimation:

$$A(s_t, a_t) = Q(s_t, a_t) - V_\phi(s_t). \qquad (9)$$

The PPO algorithm updates the actor network by maximizing the clipped surrogate objective: $L^{\text{PPO}}(\theta)$ from (5). The critic is updated to minimize the mean squared error between its predictions and the observed returns

$$L^{\text{VF}}(\phi) = \mathbb{E}_t\left[\left(V_\phi(s_t) - R_t\right)^2\right]. \qquad (10)$$

**Initialization with Off-Policy Learning.** To accelerate the initial training of EARL-BO, we employ an off-policy learning strategy to warm-start training of the RL module. As noted above, this step can use samples generated by any existing BO method. In this work, we specifically use TuRBO due to its strong performance and popularity in high-dimensional BO, though the framework could seamlessly integrate the other benchmark methods such as SAASBO, EI, etc. The off-policy learning process is as follows.

First, we generate a batch of initial training data using the chosen baseline method (TuRBO in our implementation), $\mathcal{D}_{\text{baseline}} = \{(s_i, a_i, r_i, s_i')\}_{i=1}^N$, based on the current dataset $\mathcal{D}_k$. We then use these trajectories to pre-train the actor and critic networks using a modified loss function:

$$L_{\text{pretrain}}(\theta, \phi) = \mathbb{E}_{(s,a,r,s') \sim \mathcal{D}_{\text{baseline}}}\left[L^{\text{PPO}}(\theta)\right. \\ \left. + c_1 L^{\text{VF}}(\phi) - c_2 H(\pi_\theta)\right], \qquad (11)$$

where $H(\pi_\theta)$ is the entropy of the policy, encouraging exploration, and $c_1$, $c_2$ are tunable weights. We then freeze the initial layers of both the actor and critic networks. This step preserves some pre-training knowledge, while allowing for fine-tuning in the subsequent on-policy learning phase. Specifically, the parameters are partitioned as:

$$\theta_{\text{frozen}} = \{\theta_1, ..., \theta_k\}; \quad \theta_{\text{trainable}} = \{\theta_{k+1}, ..., \theta_n\}$$
$$\phi_{\text{frozen}} = \{\phi_1, ..., \phi_k\}; \quad \phi_{\text{trainable}} = \{\phi_{k+1}, ..., \phi_n\},$$

where only $\theta_{\text{trainable}}$ and $\phi_{\text{trainable}}$ contain parameters updated in subsequent (on-policy) training. After the off-policy pre-training phase, we switch to on-policy learning using the chosen RL algorithm (PPO in our implementation).

The (optional) modular off-policy learning step provides EARL-BO with flexible initialization options, leveraging the strengths of existing BO methods while maintaining the adaptability of RL. This design choice allows practitioners to integrate EARL-BO with their existing BO infrastructure and domain-specific acquisition functions.

**Model-Based RL with GP Virtual Environment.** Typical RL algorithms learn policies through direct interaction with the environment. However, in the context of BO, where sampling is expensive, it is impractical and inefficient for an RL agent to directly interact with the real environment to find the optimal policy $\pi^*$. In these settings, it is advantageous to incorporate a (known or learned) model rather than/in addition to the environment directly, a paradigm known as *model-based* RL (Moerland et al., 2023).

To address this challenge, we propose using the current trained GP at each step as a 'model' for RL. Specifically, we create a virtual environment based on a GP fitted to the current dataset $\mathcal{D}_k = \{(x_i, y_i)\}_{i=1}^k$ and learn the optimal policy from interacting with this virtual environment as a model rather than the original black-box function.

In other words, by taking this model-based approach, EARL-BO assumes that the current GP posterior at each BO step is a good approximation of the class of functions we seek to optimize, cf. meta-learning strategies that require data from given source domains (Maraval et al., 2024). The RL algorithm is therefore provided with (multi-step lookahead) rewards using samples from the GP posterior as the underlying "black-box function" environment.

**EARL-BO Summary (Algorithm 1).** The EARL-BO algorithm implements a hybrid model-based and model-free RL approach, loosely following the Dyna framework (Silver et al., 2008; Wu et al., 2023).

The algorithm maintains a GP that serves as the world model, enabling virtual interactions for policy learning without requiring expensive function evaluations from the "real-world" environment. At each iteration, the Attention-DeepSets encoder module transforms the current state $\mathcal{D}_k$ into a latent-

---

**Algorithm 1** EARL-BO

**Input:** data $\mathcal{D}_k$, action bounds $[lb, ub]$
**Parameters:** lookahead_horizon, max_episodes, update_episodes, off_policy_episodes
**Output:** next query point $x_{t+1}$
Initialize RL agent (PPO agent), encoder network, and memory buffer
Fit GP to $\mathcal{D}_k$
**for** $k = 1$ **to** max_episodes **do**
    Reset environment state $s$ with $\mathcal{D}_k$
    **for** $step = 1$ **to** lookahead_horizon **do**
        Encode state $s$ using encoder network
        **if** $k \leq$ off_policy_episodes **then**
            Select action $a$ using TuRBO acquisition
        **else**
            Select action $a$ using RL agent
        **end if**
        Sample $\tilde{y}_{k+1} \sim \mathcal{N}\left(\mu^k(x; \mathcal{D}_k), K^k(x, x; \mathcal{D}_k)\right)$
        Compute reward $r = R(\mathcal{D}_k, x_{k+1}, \mathcal{D}_{k+1})$ using $\tilde{y}_{k+1}$
        Update environment state $s' = \mathcal{D}_{k+1}$
        Store transition $(s, a, s', r)$ to memory buffer
        $s \leftarrow s'$
    **end for**
    **if** $k \bmod$ update_episodes $= 0$ **then**
        **if** $k \leq$ off_policy_episodes **then**
            Update RL agent with initial policy
        **else**
            Calculate PPO loss using memory buffer
            Train actor, critic, and encoder networks
        **end if**
    **end if**
    Clear memory buffer
**end for**
Encode state using final encoder network
Return $x_{k+1}$ output from final actor network

---

space representation, capturing the complex relationships between previously sampled points and their corresponding values. The RL module then makes decisions based on this encoded state, and interacts with the GP model to simulate the outcomes of its actions by sampling $\tilde{y}_{k+1}$ from the GP posterior distribution.

This Dyna-style learning process allows the agent to learn from both real data (as samples are collected along the BO process) and simulated experiences (generated through GP posterior sampling). The algorithm initially learns from off-policy data generated by TuRBO (or another baseline algorithm) to establish a reasonable starting policy, then transitions to on-policy learning using PPO. As more BO samples are collected, the approximation quality of the GP model improves, and the simulated experiences become

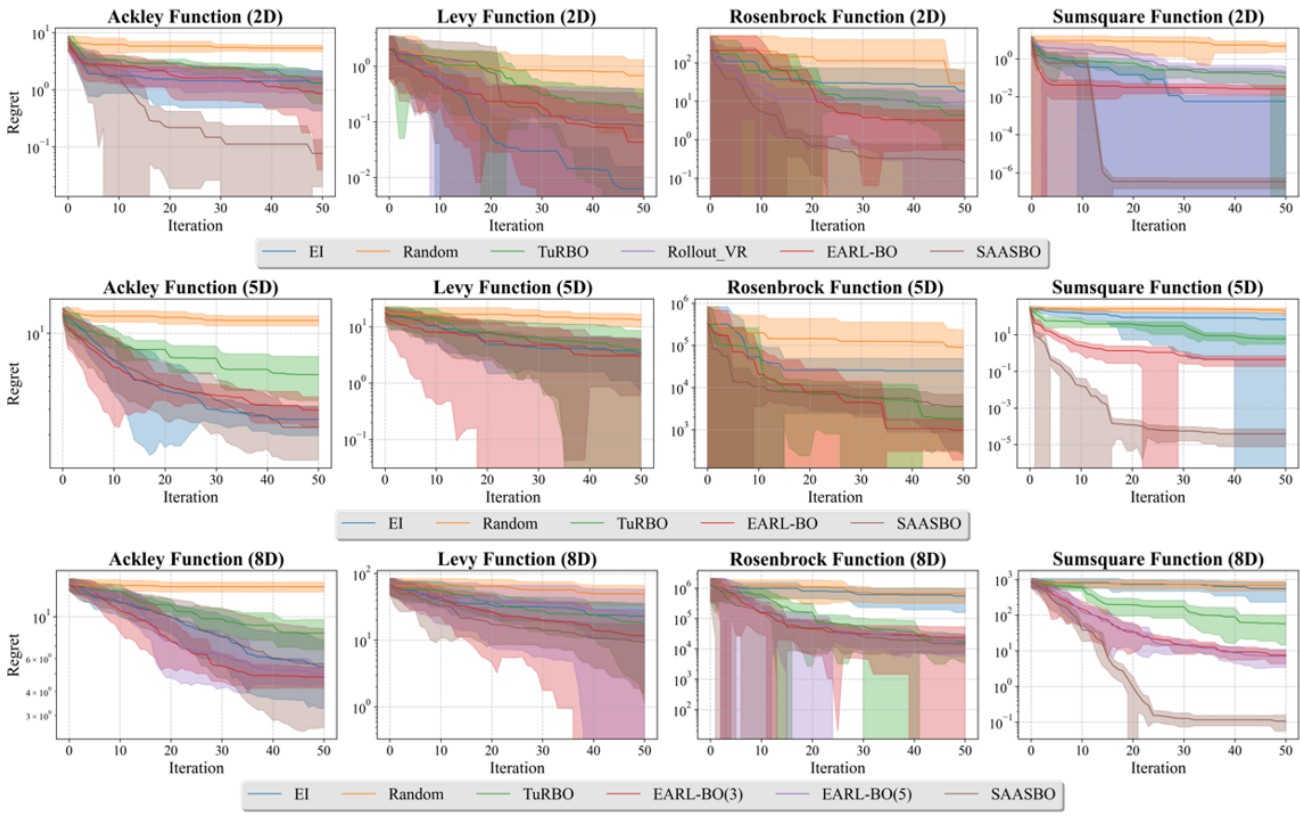

*Figure 2.* Optimization performance of various BO methods on synthetic benchmarks.

increasingly reliable for policy learning. This approach enables efficient exploration and policy improvement, while maintaining the sample efficiency that is crucial for BO.

## 4. Results and Discussion

This section compares the performance of EARL-BO against existing lookahead and high-dimensional BO algorithms on both synthetic benchmark functions and real-world hyperparameter optimization tasks. The details of the implementation and experiment are given in Appendix A. In particular, while PPO itself contains several hyperparameters, we found EARL-BO to perform well consistently and use standard/default values. The exception is the learning rate, which can be linked to the learning rate of the encoder. Appendix B contains an ablation study on learning rates.

### 4.1. Synthetic Benchmark Functions

We first evaluate EARL-BO on four popular synthetic benchmark functions: Ackley, Levy, Rosenbrock, and Sum Squares (Surjanovic & Bingham, 2013). These functions were tested in 2-D, 5-D, 8-D, and 30-D configurations, with a fixed search space of $[-15, 15]$ for all dimensions. The

optimization objective was to minimize these functions. We initialize the BO algorithms using 30 random points within the search space and evaluate performance using simple regret $(y_{opt} - y_k^*)$. Each method is tested for ten replications by resampling the initial dataset.

The results for 2-D, 5-D, and 8-D are shown in Figure 2, where the solid lines represent the mean performance across the ten runs, and the shaded area shows one standard deviation. For all problems, we compare performance against random search, standard one-step EI maximization, TuRBO, and SAASBO. For the 2-D problems, we also compare against rollout with variance reduction, denoted as 'Rollout_VR' (Lee et al., 2020). This method was impractical to run in higher dimensions (see Section 2.1). For Rollout_VR and EARL-BO, we select a lookahead horizon of $H = 3$. To investigate the effects of the lookahead horizon, we further test EARL-BO with a lookahead horizon of $H = 5$ on the 8-D benchmark functions.

The results for the 30-D setting are presented in Figure 3, using the same experimental setup (search space of [-15,15] for all dimensions, 30 initial data points) as the lower-dimensional experiments. For a better visibility of the figure, we excluded random search from the 30-D comparisons,

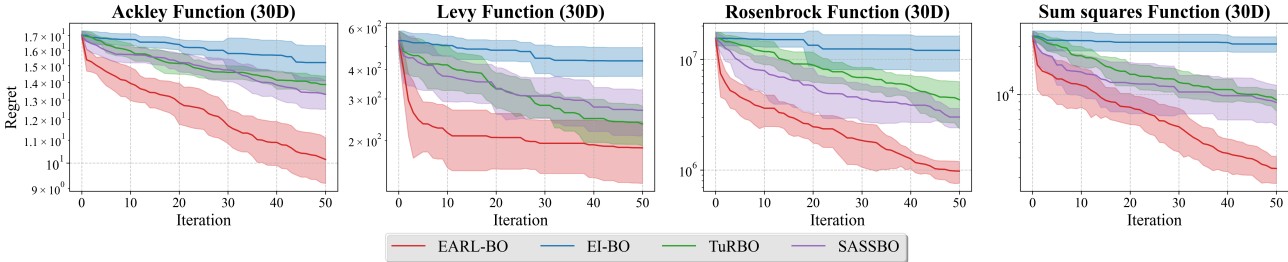

*Figure 3.* Optimization performance of various BO methods on 30-D synthetic benchmarks.

focusing on the more competitive BO methods.

**Results.** As shown in Figure 2, in the 2-D setting, EARL-BO demonstrates competitive performance, often closely matching, or slightly improving on, the performance of Rollout_VR. This alignment is intuitive, as both methods implement multi-step lookahead strategies, suggesting that EARL-BO is able to properly learn the three-step lookahead policy. This is notable given the relative computational scalability of EARL-BO, as described in Appendix C. For most functions, the performance gap between EARL-BO and single-step lookahead methods such as EI was not as pronounced in this low-dimensional setting, likely due to the relatively high information density provided by the 30 initial points in a 2-D space, which may reduce the advantages gained from extended exploration and lookahead strategies. In other words, myopic policies are less detrimental since fewer samples are required.

A notable pattern emerges with the Sum Squares function across all dimensionalities (2-D, 5-D, and 8-D), where SAASBO consistently demonstrates superior performance compared to all other methods, including EARL-BO. This result is also intuitive, given the additive nature of the black-box objective. The performance highlights SAASBO's particular effectiveness on smooth, unimodal functions, where its sparse axis-aligned approach can efficiently navigate the search space. However, for functions with more complicated landscapes featuring multiple local optima or irregular surfaces, the relative performance changes significantly.

In the 5-D and 8-D settings, EARL-BO's advantages become increasingly evident on the more complex functions. Across the Ackley, Levy, and Rosenbrock functions, EARL-BO consistently outperforms baseline methods including EI, Random Search, and TuRBO. This trend underscores the effectiveness of our multi-step lookahead approach in navigating more complicated functions and higher-dimensional landscapes (relative to other lookahead methods).

The 30-D experimental results, shown in Figure 3, demonstrate EARL-BO's outstanding performance in high-dimensional optimization settings. Across all four 30-D benchmark functions, EARL-BO significantly outperforms EI, TuRBO, and SAASBO, with performance gaps that are substantially larger than those observed in lower dimensions. This pronounced improvement supports our hypothesis that, as problem dimensionality increases, the optimization landscape becomes more complex and challenging, making non-myopic decision-making increasingly valuable. The ability to plan multiple steps ahead becomes crucial for navigating high-dimensional spaces effectively, where myopic policies are more likely to become trapped in suboptimal regions or fail to exploit promising areas of the search space efficiently.

Notably, even SAASBO, which performs competitively on certain lower-dimensional problems such as Sum Squares, is consistently outperformed by EARL-BO in the 30-D setting across all benchmark functions. This suggests that the benefits of multi-step lookahead planning become dominant over specialized high-dimensional techniques as the complexity of the optimization problem increases.

Regarding lookahead horizons, the three-step lookahead variant of EARL-BO shows similar or slightly better performance compared to its five-step counterpart in the 8-D experiments. This finding aligns with empirical observations from previous multi-step lookahead studies (Lam et al., 2016; Lee et al., 2020), which report optimal performance at intermediate horizons of $H = 3, 4$ steps compared to $H = 5, 6$. A more comprehensive study on more lookahead horizon lengths can be found in Appendix B.

The enhanced performance of EARL-BO in higher dimensions may be attributed to several factors:

- **Increased exploration benefits:** For complex and higher-dimensional functions, the ability to plan multiple steps ahead becomes more valuable for effective exploration and exploitation balance.

- **Efficient use of information:** EARL-BO's Attention-DeepSets encoder better captures and exploits relationships between higher-dimensional samples, enabling more informed decision-making.

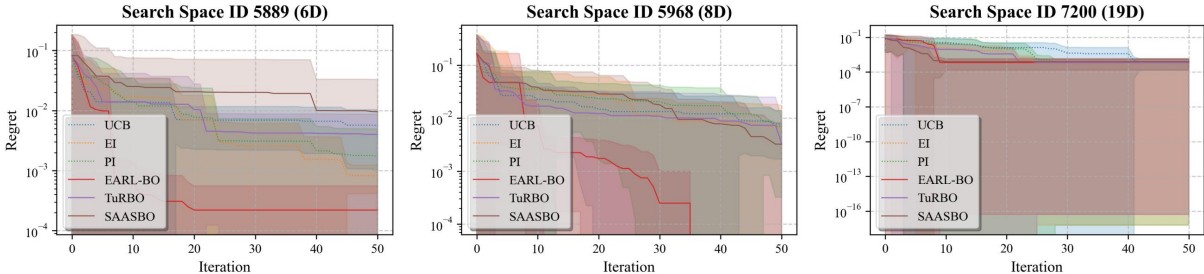

*Figure 4.* Optimization performance of various BO methods on HPO problems.

In addition to the aforementioned heuristic observations, the plateauing, or slight degradation in performance with increased lookahead (from three to five in 8-D), may also be attributed to errors in the GP model, a phenomenon we term "planning delusion." As EARL-BO does not interact with the true objective function, but rather with samples from the GP model (which then interacts with the true objective function), looking many steps ahead may compound errors in the predicted outcomes. In essence, the true objective function may not correspond exactly to a GP posterior sample, reducing the effectiveness of model-based learning. The three-step variant seemingly allows for meaningful planning without excessive reliance on imperfect models.

We speculate that this "planning delusion" effect could be more pronounced in EARL-BO compared to traditional rollout-based methods. EARL-BO learns a policy that directly optimizes for multi-step performance, potentially making it more susceptible to biases in long-term predictions. In contrast, rollout methods typically use myopic policies as base strategies, which may provide some inherent robustness against long-horizon planning errors.

Further results on the synthetic benchmarks, including ablation studies on learning rates, planning delusion, RL training convergence, and permutation invariance, are given in Appendix B. Computational requirements and scalability are discussed in Appendix C.

### 4.2. Hyperparameter Optimization Experiments

We next evaluate EARL-BO in real-world scenarios using the Hyperparameter Optimization Benchmarks (HPO-B) dataset (Arango et al., 2021). HPO-B contains datasets of classification model hyperparameters and their corresponding accuracies across multiple types and search spaces, providing a realistic testbed for optimization algorithms. We focus on higher-dimensional optimization problems, and select search space IDs 5889 (6-D), 5968 (8-D), and 7200 (19-D) as test problems. We initialize the BO algorithms using five random points for 6- and 8-D and 50 for the 19-D problem. We again evaluate performance using simple regret $(y_{opt} - y_k^*)$ and ten replications by resampling the initial

dataset. We compare performance against random search, standard EI and PI maximization, TuRBO, and SAASBO. For EARL-BO, we select a lookahead horizon of $H = 3$.

**Results.** As shown in Figure 4, EARL-BO consistently outperforms the baseline methods on the HPO tasks. Appendix B visualizes these results with non-logarithmic scaling and contains an ablation experiment for the 19-D problem starting from only five initial samples. For all problems, EARL-BO performs equivalently (or worse than) competing methods at early iterations, and then overtakes them quickly later, including TuRBO and SAASBO. This suggests that single-step methods are indeed overly myopic, and thus focused on initial rewards; the multi-step lookahead enables EARL-BO to achieve the best long-term performance.

In the 19-D Search Space (ID 7200) setting with only five initial points (Appendix B), EARL-BO performs similarly to some of the baseline methods. In this case, the initial data are extremely sparse, leading to a poor GP model initially, and as a result, large variation in samples from the GP posterior. In this scenario, EARL-BO suffers again from model errors, and adopting its multi-step lookahead policy may not provide the expected benefits and could potentially lead to suboptimal decisions due to errors in long-term predictions.

## 5. Conclusions

This paper introduces EARL-BO, a novel Bayesian Optimization approach based on Encoder-Augmented Reinforcement Learning for multi-step lookahead in high-dimensional spaces. The approach combines an Attention-Deepsets encoder module with model-based, actor-critic RL to provide an end-to-end solution for the sequential decision-making process of BO. Our experiments on synthetic benchmarks and hyperparameter optimization tasks demonstrate EARL-BO's superior performance in moderate to high-dimensional spaces compared to traditional and state-of-the-art methods. Overall, this study demonstrates the potential of reinforcement learning in handling the sequential nature of BO.

# Acknowledgements

The authors thank Dr Mark van der Wilk for discussions and insightful feedback, especially in the conceptualization of this project. CT gratefully acknowledges support from a BASF/Royal Academy of Engineering Senior Research Fellowship. This work was partially supported by the Basic Science research program through the National Research Foundation of Korea (NRF) funded by the Ministry of Science, ICT & Future Planning under the contract No.NRF-2021R1C1C1012014 as well as USC ECET's funding under CYPRES award as part of the distribution of a settlement relating to fuel economy for gasoline-powered vehicles.

# Impact Statement

This paper presents work whose goal is to advance the field of Machine Learning through improved Bayesian optimization techniques. The potential societal consequences align with those generally associated with advancing machine learning methodologies.

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

# A. Experimental Setup

In this section, we provide details of the experimental setup, focusing on the dataset, its preprocessing, the hyperparameters of RL, and the hardware used for experiments to aid in reproducibility.

## A.1. Tasks

The real-world Hyperparameter Optimization dataset is sourced from the HPO-B dataset (Arango et al., 2021), a collection of HPO datasets grouped by search space and tasks. Each search space ID refers to the set of hyperparameters for a specific machine learning (ML) model, e.g. Ranger, XGBoost. Table 5 of (Arango et al., 2021) gives the details for each search space ID, including number of evaluations, number of datasets, number of dimensions, and the name of search space (i.e., the name of ML model). Depending on the number of combined datasets, some search spaces contain duplicate measurements (i.e. multiple/different response recorded for the exact same input values). To mitigate this issue, (Arango et al., 2021) randomly select one data sample for each input when duplicates exist; however, in this research, we only use search space IDs which do not contain data duplication for cleaner BO comparisons. All in all, due to our research topic in 'high-dimensional' problems and for the clean-data, we have selected search space IDs of 5889 (6-D), 5968 (8-D), and 7200 (19-D).

## A.2. Baselines

For baseline optimization performance on the synthetic benchmark functions, we compare against EI, Random, TuRBO, and Rollout_VR in this work. For EI, Random, and Rollout_VR, we use implementations from (Lee et al., 2020) found at: `https://github.com/erichanslee/lookahead_release`. For TuRBO, we use the current implementation from the Uber research group: `https://github.com/uber-research/TuRBO`. To keep experiments consistent and reproducible, optimization was performed in the same search space with same initial data for each problem repetition. Note that we found TuRBO can give slightly different optimization performance even with the same set of initial data. Ten sets of initial data were provided to optimization methods, and the average performance is reported in this paper.

## A.3. EARL-BO

On top of the pseudo code (i.e., Algorithm 1) presented in the main text, EARL-BO contains one additional detail—namely the stopping criterion for RL training. For this, if the PPO agent fails to learn policy over some episodes (e.g., average reward becomes zero for 15 consecutive updates) or the final trained agent's average reward is less than 1e-5, EARL-BO aborts the learned policy and returns the off-policy suggestion provided by the TuRBO algorithm. We found this to happen in practice only seldom, particularly when EARL-BO had already discovered the optimum point.

**Hyperparameters.** Table 1 displays a comprehensive list of hyperparameters for EARL-BO. We would like to underline that none of these presented values for hyperparameters were tuned across problems. In other words, across various dimensions and function forms, we have kept the same hyperparameters with most basic PPO and Encoder values. However, setting learning rates for the RL and encoder modules (their relative magnitudes in particular) was a very important question. We employed different learning rates for the RL agent (0.001) and the encoder (0.01). This design choice is justified by the distinct roles and complexities of these components within the algorithm, allowing them to train in a dynamically decoupled fashion. The RL agent, tasked with learning a policy and value function in a dynamic environment, benefits from a lower learning rate to maintain stability and prevent performance collapse due to rapid policy changes. Conversely, the encoder, which learns a static representation of the search space, can adapt more quickly with a higher learning rate, improving the quality of state representations rapidly. As an ablation study below, we present EARL-BO with 4 different selections of learning rates and compare the optimization performance.

## A.4. Hardware

We conducted our experiments on a computing server with AMD EPYC 7742 processors equipped. The specific allocation for each job was as follows: 16 CPUs and max memory of 100 GB. With this configuration, the average time required to complete each experiment was approximately 25 hours. This configuration was chosen in order to parallelize the multiple repetitions for each experiment. Notably, experiments could potentially be sped up using GPU acceleration.

*Table 1.* EARL-BO hyperparameter values.

| PPO | |
|---|---|
| Learning rate | 0.001 |
| # epochs | 100 |
| Epsilon clip $\epsilon$ | 0.2 |
| $\beta$ values for Adam | (0.9, 0.999) |
| Discount factor $\gamma$ | 0.95 |
| Value function coefficient | 0.5 |
| Entropy coefficient | 0.1 |
| # layers frozen | 2 |
| Max episodes | 4000 |
| Update frequency | 50 |
| # off-policy episodes | 400 |
| No-improvement threshold | 15 |
| Horizon | 5 |
| **Encoder** | |
| Hidden dimension | 64 |
| Output dimension | 16 |
| Learning rate | 0.01 |
| **GP** | |
| Kernel | RBF + WhiteKernel |
| RBF length-scale bounds | (1e-2, 1e2) |
| Noise bounds | (1e-10, 1e1) |

# B. Additional Results

In this section, we describe several ablation studies to characterize the proposed EARL-BO algorithm. Specifically, should the relative learning rate of RL be slower than that of the encoder? Does "planning delusion" occur only when the lookahead horizon is long? How stable is the RL training across BO iterations? Is permutation invariance important?

**What is the effect of different learning rates?** Figure A1 shows two different cases for 8-D optimization of the Ackley function: first, when the learning rates for the RL and encoder modules are, respectively, (0.001, 0.01). This configuration is denoted as 'standard.' We also reverse the learning rates to be (0.01, 0.001), respectively, which is indicated as 'reverse.' Figure A1 shows that having a higher learning rate for the RL module leads to both unstable BO performance (as we can observe the increasing standard deviation over time), but also lower optimization performance.

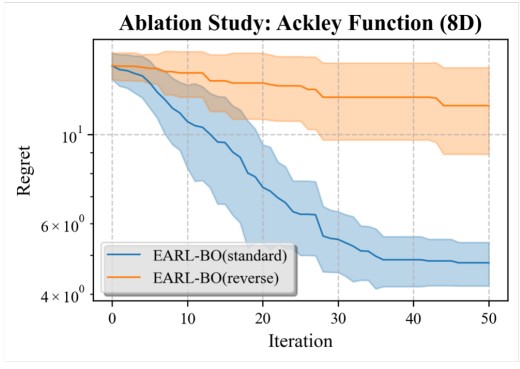

*Figure A1.* Optimization performance of EARL-BO with different relative learning rates.

**When can planning delusion happen?** In the Results and Discussion section of the main text, we discuss that "planning delusion" may happen if the lookahead horizon for RL is large, due to high uncertainty of the GP virtual environment. Astute readers will notice that, if the high-uncertainty of the GP is the source of this planning delusion, having extremely scarce data for high-dimensional optimization might also cause the planning delusion even when the lookahead horizon

is small. Figure A2 repeats the HPO-B search space ID 7200 (19-D) problem with only five initial samples instead of 50. The results confirm that, if we employ EARL-BO in high-dimensional search spaces with sparse initial data, it may indeed underperform compared to other BO methods, suffering from planning delusion.

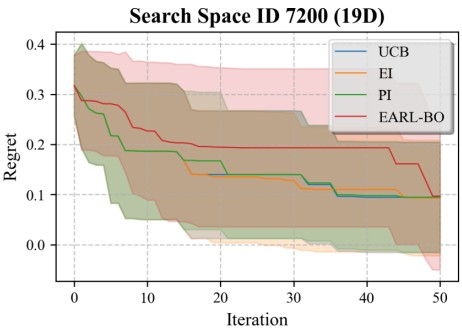

*Figure A2.* Optimization performance of methods on HPO-B search space ID 7200 with 5 initial samples.

**Additional Figures.** Figure 3 in the main text might suggest that the standard deviations of performance do not decline as BO progresses for the HPO-B dataset. This can be attributed to the logarithmic y-axis scale in Figure 3. Thus, we include here the same graph with a linear y-axis scale. Figure A3 more easily visualizes that the standard deviation of regrets in all the BO methods are decreasing over as BO progresses. More specifically, in most of cases, EARL-BO displays the smallest standard deviation after approximately eight iterations among all compared BO methods. This suggests that EARL-BO exhibits relatively stable performance regardless of initial point distribution.

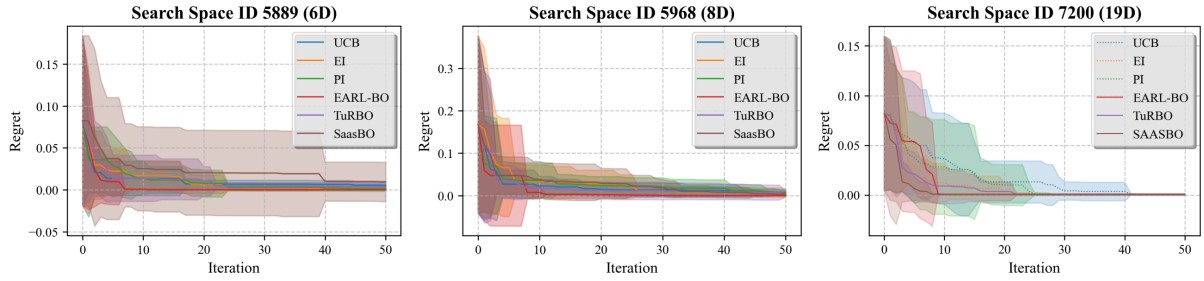

*Figure A3.* Performance of various BO methods on benchmark functions with non-logarithmic y-axis.

**How stable is the RL training across BO iterations?** Figure A4 presents the RL training curves at several BO steps (1st, 11th, and 21st) during the optimization of the 8-D benchmark functions. It suggests two possible characteristics of EARL-BO. First, the RL training appears stable across all BO iterations, as evidenced by the smooth convergence of the loss function. Second, we observe that the RL training may become slightly easier as BO progresses–the loss functions for the 11th and 21st BO steps start at and converge to lower values compared to the first step. This suggests that as EARL-BO accumulates more information about the optimization landscape, the RL agent may more easily learn effective decision-making policies, possibly also due to the improved quality of the learned encoded representations.

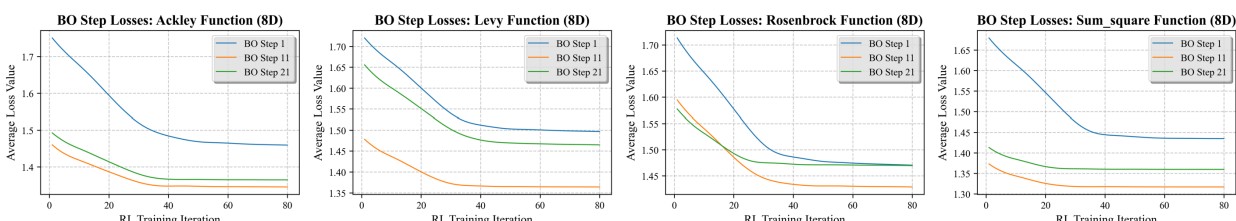

*Figure A4.* Convergence of RL training at various BO iterations.

**Is permutation invariance important?** Figure A5 presents an ablation study investigating the importance of permutation invariance in the encoder design. We compare EARL-BO with two encoder architectures on the 8D Ackley function:

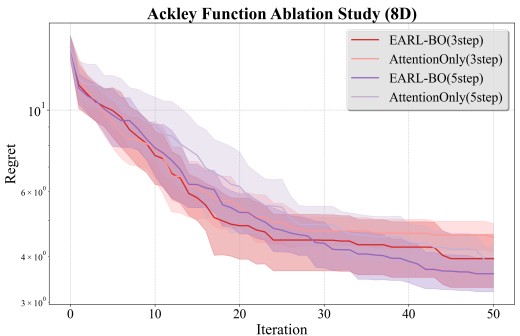

*Figure A5.* Performance of EARL-BO with different encoders.

The two tested architectures are both attention-based, but differ in that one includes the DeepSets architecture for permutation invariance, while the other is 'attention-only.' We evaluate both designs using three- and five-step lookahead horizons. The results show that EARL-BO with the Attention-DeepSets encoder consistently outperforms the attention-only variant across both lookahead settings. This validates our architectural choice and confirms that explicit permutation invariance is beneficial for BO decision-making, intuitive since the order of data acquisition should not affect the learning process.

**Effect of different lookahead horizons** Figure A6 shows the optimization performance of EARL-BO across different lookahead horizons (1, 3, 5, and 7 steps) on the 8D Ackley function.

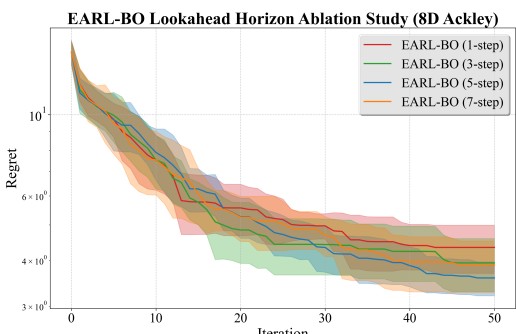

*Figure A6.* Performance of EARL-BO with different lookahead horizons.

The one-step lookahead variant demonstrates strong initial performance but exhibits the worst optimization results by the end, consistent with the intuition that myopic behavior prioritizes immediate gains over long-term exploration. The three-step and five-step lookahead variants achieve the best overall performance. The seven-step variant exhibits generally poor performance throughout most iterations, though with some improvement toward the end. These results align with empirical observations from previous multi-step lookahead studies (Lam et al., 2016; Lee et al., 2020), which report optimal performance using intermediate lengths for lookahead horizon.

## C. Computational Cost Analysis

While EARL-BO demonstrates superior optimization performance on most test problems, computational efficiency represents an important practical consideration. In this section, we provide a detailed analysis of the computational overheads associated with different BO methods.

### C.1. Runtime Comparison on 8D Ackley Function

Table 2 presents the computational cost results for various BO methods on the 8D Ackley function. All given times represent the average runtime (i.e., time to compute the next sample location) per BO iteration across 10 independent runs.

*Table 2.* Computational cost comparison on 8D Ackley function.

| Method | Average Runtime (s) | Std Dev (s) | Notes |
|---|---|---|---|
| EI | 0.28 | 0.05 | – |
| TuRBO | 0.27 | 0.20 | – |
| SAASBO | 168.6 | 27.6 | – |
| Rollout_EI (3-step) | >3600 | – | 1000 MC iterations |
| EARL-BO (3-step) | 840 | 147 | 4000 episodes |
| EARL-BO (5-step) | 1075 | 134 | 4000 episodes |

These computational cost results reveal several important insights:

**Myopic vs. non-myopic methods:** As expected, myopic methods (EI, TuRBO) exhibit significantly faster per-iteration computation times compared to multi-step lookahead approaches. However, SAASBO, despite being a myopic method, requires substantially more computation time due to its sparse axis-aligned subspace optimization.

**EARL-BO vs. rollout methods:** EARL-BO achieves multi-step lookahead performance with considerably less computational overhead than traditional rollout-based methods. The rollout-based method with variance reduction exceeds our 3600-second timeout threshold even with only 1000 Monte Carlo iterations, while EARL-BO completes its 3-step lookahead training in approximately 840 seconds. Note that significantly more MC iterations may be required for good performance in higher dimensional settings or longer lookahead horizons.

**Horizon effects:** The computational cost scales moderately with increased lookahead horizon (cf. rollout), with 5-step EARL-BO requiring only 28% more time than the 3-step variant.

### C.2. Scalability to Higher Dimensions

The results on the 30-dimensional Ackley function further demonstrate the computational scalability of EARL-BO. In particular, the computational costs here highlight the relative efficiency of our framework:

- **Rollout methods** would require exponentially more Monte Carlo iterations to handle the expanded search space, making them computationally prohibitive.

- **EARL-BO** runs in ∼1600 seconds for 30D problems, representing only a 100% increase compared to 8D optimization.

- **Myopic methods** EI and TuRBO require 533% and 215% more CPU time, respectively, when scaling from 8D to 30D.

These results demonstrate that EARL-BO's computational scaling is more favorable than both traditional rollout methods and some myopic approaches when moving to higher-dimensional problems.

### C.3. Practical Considerations

While multi-step lookahead methods inevitably require more computation time compared to myopic approaches, several factors justify this overhead in practical BO applications:

**Expensive function evaluations:** BO is typically applied to problems where individual function evaluations are costly (e.g., materials discovery, hyperparameter tuning of large models, chemical laboratory experiments). In these settings, a single point evaluation can require hours or days, making the BO overhead relatively insignificant.

**Scalability advantages:** Unlike rollout-based methods that become computationally intractable in higher dimensions, EARL-BO maintains reasonable computational requirements while scaling to practically relevant dimensionalities.

The computational analysis confirms that while EARL-BO introduces additional overhead compared to myopic methods, its superior optimization performance and favorable scaling properties make it particularly suitable for expensive black-box optimization problems where the evaluation costs dominate the total optimization budget.

