# OpenReview forum: "EARL-BO: Reinforcement Learning for Multi-Step Lookahead, High-Dimensional Bayesian Optimization"
_ICML.cc/2025/Conference — ICML 2025 poster_

### Official Review · Reviewer_Kq9J · 2025-03-12

**Overall Recommendation:** 2

**Summary:**

This paper propose a quite interesting and promising way to combine strength of RL with effective BO optimizer. The authors first pose the limitations of current BO optimizer and hence locate the "myoptic" issue wihtin these BO methods. To address that, the authors propose modelling the BO optimization iterations as MDP~(at least POMDP) and learn a sampling neural network by k-step PPO. A series of novel designs have been proposed by the authors to facilitate this technical objective. First, the authors give a reasonable MDP definition that align well with BO optimziation process, including the state reprsentation, action, max(.,.) reward and transition dynamics. With the MDP definition, an attention-based state extraction mechanism is introduced  with a weighted scoring function, which could help extract order-invariant and size-invariant information from gradually enlarged query points set D. The RL agent in this paper serve as a sampling method to sample next query point and the reward function rewards actions with maximum EI value. Moreover, the authors carefully designed the training procedure, which include two stages: 1) off-policy learning from advanced BO method; 2) further on-policy fine-tuning with parts of the network parameters. The effectiveness of the learned BO policy is tested on both synthetic and reaslistic optimziation scenarios and shows improved performance against several baselines.

## after the rebuttal

Thanks for the responses. Overall, I still think the added validation and further explanation are not convincing enough. I keep my boardline rejection opinion on this paper. However, I respect the other two reviewers and AC/PC and would leave the final decision to them.
The writing of this paper is clear and well organized.

**Claims And Evidence:**

I acknowledge and agree one of the authors:

a) BO has myoptic issue and using MDP modelling and RL for solving the MDP could address this issue. This claim is clearly supported by the elaboration of motivation, related works and methodology in Section 2 and Section 3.

There are three claims that I think are not fairly validated and supported:

a) In page 2, left column, line 073, the authors claim that "An integrated RL-based BO method that efficiently solves the SDP inherent in multi-step BO". However, no empirical evidence is provided to demonstrate that point. Is the efficiency here means the computational complexity? If so, this method proposed in this paper involve RL training, isn't is less efficient than traditional BOs?

b) The authors claim that the proposed method aims to address high-dimensioanl problems. However, the problem with maximum dimensions in the experiment is 19-D, which is apparently not a high-dimensional problem.

c) In page 4, right column, line 205-208, the authors claim that "Permutation invariance is crucial in BO, as the order in which data points
is acquired should not affect the learning process". I wonder whether this claim is correct since the sampling trajectories show clear time-dependent features as the GP approximation transforms from a state to another. I strongly request the authors to add an ablation where position encodings are added to the sampling point trajectories and compare with the current version. This would definitily help back up this claim you have posed and seriously affect the correctness of your neural network designs.

**Essential References Not Discussed:**

As I said above, since the topic discussed in this paper falls into Learning to Optimize domain, at least some latest development within this scope should be cited in the beginning of the paper. In particular, since this paper focus on continuous black-box optimization domain, I suggest the following related papers:

[1] https://dl.acm.org/doi/abs/10.1145/3321707.3321813

[2] https://openreview.net/forum?id=h3jZCLjhtmV

[3] https://arxiv.org/abs/2402.17423

[4] https://ui.adsabs.harvard.edu/abs/2024arXiv240503728L/abstract

[5] https://arxiv.org/abs/2411.00625

**Experimental Designs Or Analyses:**

The experimental protocols are somehow insufficient:

a) At least one evolution strategy baseline should be compared since ES also show robust optimziation performance on expensive and high-dimensional problems. I give two suggestions: Sep-CMAES (https://link.springer.com/chapter/10.1007/978-3-540-87700-4_30) and MMES (https://arxiv.org/pdf/2203.12675).

b) The authors only present three problem instances in HPO-B, which makes me wonder the perforamnce of the propsoed method on many other HPO isntances in the benchmark. More instances should be involved and average performance and error bars are needed to demonstrate the efrectiveness. Besides, instances in HPO-B can not fully represent challenging high-dimensional optimziation problems, consider NeuroEvolution benchmarks.

c) There are no ablation studies to explore the true effectiveness of the proposed designs.

**Methods And Evaluation Criteria:**

N/A

**Other Comments Or Suggestions:**

a) I suggest shorten the lengthy text content about related works in Section 2.1, about RL and PPO algorithms in Section 3.1 and Section 3.2. Instead, provide sufficient ablation study results to reinforce the scientific integrity of this paper.

b) The begining of Section 2, there are many "K" symbols, with different meanings. Correct them please.

c) Figure 1 is comprehensive yet might be too comprehensive to understand the concept of the propsoed work properly. Simplifying this figure would help readers indeed.

d) a small question, in page 5, left column, line 232-233, the authors claim that an arithmetic mean could also be used as aggregation function. However, it might reduces the bias hence makes it hard for RL to learn, isn't it? The design now, which is a neural network could provide sufficient bias, from my opinion.

**Other Strengths And Weaknesses:**

N/A

**Questions For Authors:**

See my comments in other blocks.

**Relation To Broader Scientific Literature:**

I notice that this paper show relation with a broader research domain: Learning to Optimize (L2O). Within this scope, the researches widely dicuss how learning techniques such as reinfircement learning can be leveraged to boost traditional optimization methods.

**Theoretical Claims:**

There are no theoretical claims in this paper.

---

> ### Author Rebuttal · Authors · 2025-04-01
>
> We sincerely thank the reviewer for their detailed assessment of our work and for raising important questions that help strengthen the paper. We appreciate the time taken to thoroughly review our submission and provide constructive feedback.
>
> ### **Regarding the claim about "efficiently solving SDP":**
>
> We apologize for the lack of clarity in our terminology. Our claim here refers to the relative efficiency of RL in (approximately) solving SDPs, compared to other *multi-step lookahead* BO methods, as the traditional single-step BO approaches do not solve the SDP. For example, our 3-step lookahead EARL-BO demonstrates superior optimization performance across various benchmark functions compared to the Rollout_BO, while also having smaller computational time requirements. Efficiency is indeed an important aspect, which we will explore in a revised version. Please see the response to Reviewer eu5H for our preliminary analysis of CPU time for each BO method on the 8D Ackley function.
>
> ### **On addressing "high-dimensional problems":**
>
> We should clarify that we tried to emphasize in our writing that by "high-dimensional," we mean higher-dimensional again in comparison to other *non-myopic* BO methods, which typically handle only very low-dimensional problems (≤6D). EARL-BO successfully scales to practically relevant dimensionalities (up to 19D in our experiments), which represents an advancement for non-myopic BO approaches. Moreover, EARL-BO is a modular framework, and scalability depends mostly on the chosen RL method, providing avenues for future improvement. We will revise our claims in the introduction to be more precise about this contribution.
>
> ### **Regarding the ablation study on permutation invariance:**
>
> We appreciate this excellent suggestion for understanding not only the architecture of EARL-BO, but also the general permutation invariance of Bayesian optimization as an SDP. Following your recommendation, we will include an ablation study comparing an attention-only encoder (with position encodings) to our attention-DeepSets based encoder to exactly explore this point. We have preliminarily run this experiment on the 8D Ackley function for both 3-step and 5-step lookahead settings. Results can be found here: https://anonymous.4open.science/r/icml_2025_review-B587. The results confirm that our proposed method achieves significantly better optimization performance across different BO settings compared to the attention-only encoder, supporting our claims about the importance of permutation invariance. We will include these findings, which support our EARL-BO framework, in the revised version.
>
> ### **On evolution strategy baselines:**
>
> We thank the reviewer for suggesting additional baselines. We want to clarify that our primary aim is to extend the capabilities of non-myopic Bayesian optimization, rather than to compete with methods designed for very high-dimensional problems (100s-1000s of dimensions), and thus large numbers of samples (on the order of 10^6 in the provided references). We will clarify our contributions in the introduction and reference these evolution strategies and Learning to Optimize (L2O) as alternative approaches to BO for different problem settings. This important distinction will help position our work more accurately within the literature.
>
> ### **Regarding the minor point about aggregation functions:**
>
> This comment raises an important point about the potential trade-off between bias and learnability when choosing aggregation functions. While we mentioned arithmetic mean as a permutation-invariant alternative to summation, we agree that maintaining an appropriate level of inductive bias is crucial for effective RL training. Our neural network-based approach provides sufficient bias to facilitate learning, as you noted. We will clarify this discussion in the revised manuscript.
>
> ### **Additional improvements:**
>
> We thank the reviewer for the detailed feedback and will address the other suggestions in a revised version, including: fixing the inconsistent "K" symbols in Section 2, correcting the directory titles in the supplementary source code, improving clarity of Figure 1, and expanding our experimental evaluations with the ablation studies suggested throughout the reviews.
>
> We sincerely appreciate the reviewer's constructive feedback, which has helped us identify areas for improvement and clarification. We believe these changes will strengthen our paper significantly.

---

> > ### Comment · Reviewer_Kq9J · 2025-04-03
> >
> > I am generally satisfied with some responses of the authors.
> >
> > I still have following concerns:
> >
> > I have read all review comments (the other two reviewers and mine) and the corresponding responses from the authors. Especially, I found the computatioanl efficiency issue raised by reviewer euH5 is fatal for this paper, since the problem dimension this work could address is less than 20, while the solving time is so huge.'
> >
> > Regarding the ablation study on permutation invariance, the additional experimental results (https://anonymous.4open.science/r/icml_2025_review-B587) show very close performance with overlapping error bars, how could this be a significant demonstration?

---

> > > ### Author Response · Authors · 2025-04-08
> > >
> > > Thank you for acknowledging our efforts and for the continued engagement.
> > >
> > > ### [Computational Efficiency]
> > > Regarding computational efficiency comparisons, please see our discussions with Reviewer eu5H. We reproduce from that discussion a key point on computational time here:
> > >
> > > Despite strongly outperforming rollout-based methods, we acknowledge the reviewers’ concerns regarding computational efficiency compared to the cheaper (myopic) methods. We do acknowledge this limitation of EARL-BO (and non-myopic methods in general), but note that BO is specifically designed for optimization problems where function evaluations are expensive, such as hyperparameter tuning for ML models [1], engineering design/simulations [2], and chemical laboratory experiments [3]. In these settings, a single function evaluation can take hours or even days, not to mention monetary costs, while the overhead of BO remains relatively small (difficulty of these problems is also mentioned by Reviewer JCQV). Thus, even as EARL-BO introduces additional computational cost relative to *myopic* baseline BO approaches, its significant improvement in performance can justify this expense. Furthermore, the improved sample efficiency of our method means that fewer function evaluations are required to reach an optimal or near-optimal solution. This results in a (significant) net reduction in total cost when considering the entire optimization process. Future works can exploit the modularity of EARL-BO for faster settings, e.g., by warm-starting RL, alternative RL methods/tunings, using the actor for multiple BO iterations, etc.
> > >
> > > ### [Permutation Invariance]
> > > We agree that the performance gap between the DeepSets and Attention-Only frameworks is limited. Nevertheless, despite the standard deviations in this preliminary study we do already see a *consistent* performance improvement using DeepSets (noting the logarithmic y-axis): after 10 iterations, mean regret is 7.8 vs 9.1; after 20 iterations, 5.3 vs 6.2; after 30 iterations, 4.3 vs 4.9; after 40 iterations, 3.9 vs 4.4; and after 50 iterations, 3.6 vs 3.9. These slight, but consistent improvements are non-trivial in expensive settings, e.g., see our discussion above.
> > >
> > > While we hope to have time to produce more comprehensive results on this study in a revised version, these preliminary results already reveal some interesting conclusions: (1) these results begin to answer ‘*How Markovian is BO?*’. This is particularly interesting given that EARL-BO may exhibit some *synthetic* non-Markovian behavior, as the samples can become less reliable further into the lookahead horizon (see discussions on planning delusion with Reviewers euH5 and JCQV); (2) practically speaking, noting the modular design of EARL-BO, these results may suggest improvements for the encoder design in future works.
> > >
> > > We believe these are highly interesting areas to discuss in a revised version and are grateful to the Reviewer for suggesting this ablation study and direction of investigation.
> > >
> > > ### [Scalability of EARL-BO]
> > > To further highlight the relative efficiency of the EARL-BO framework, we will include optimization results on the 30-dimensional Ackley function. A preliminary version of this experiment (two random starts instead of ten for now) can be found at the same anonymous repository https://anonymous.4open.science/r/icml_2025_review-B587. EARL-BO significantly outperforms all comparison methods, again demonstrating its scalability to challenging optimization problems in practice.
> > >
> > > The computational requirements for this added experiment emphasize our scalability results: the rollout-based method would require even more Monte Carlo iterations to handle this search space, yet using even 1000 iterations already times out in the 3600s budget. On the other hand, EARL-BO runs in approximately 1600s, a moderate increase in time compared to the much less complicated setting of optimization over the 8-D Ackley function. Specifically, compared to 8-D 3-step lookahead EARL-BO, BO iterations in the 30-D problem require approximately 100% added CPU time, while myopic methods, such as EI and TuRBO require 533% and 215% more CPU time, respectively, to make decisions in 30-D compared to the 8-D optimization problem.
> > >
> > > We believe these results strengthen our contribution and will include a more comprehensive analysis in a revised version.
> > >
> > > **References:**
> > >
> > > [1] Snoek, J., Larochelle, H., & Adams, R. P. (2012). Practical Bayesian optimization of machine learning algorithms. NeurIPS 2012.
> > >
> > > [2] Jones, D. R., Schonlau, M., & Welch, W. J. (1998). Efficient global optimization of expensive black-box functions. Journal of Global Optimization, 13, 455-492.
> > >
> > > [3] Shields, B. J., Stevens, J., Li, J., Parasram, M., Damani, F., Alvarado, J. I. M., Janey, J. M., Adams, R. P., & Doyle, A. G. (2021). Bayesian reaction optimization as a tool for chemical synthesis. Nature, 590(7844), 89-96.

---

### Official Review · Reviewer_JCQV · 2025-03-13

**Overall Recommendation:** 4

**Summary:**

The paper proposes a new framework for combining RL (via meta-learning) with BO to provide effective optimisation. They propose a NN architecture that deals well with the sequential nature of data-collecting. For each BO step, off-line policy learning attempts to learn how to mimic TURBO, and then model-based RL is used to fine-tune the policy. Unlike exisiting RL-based optimisers, the RL agent is trained fro scratch from each BO step allowing a higher level of specificity. In particular, they focus on the task of non-myopic optimisation, providing empirical results.

## update after rebuttal

Thanks for your convincing replies to the rebuttal. I maintain my already high score.

**Claims And Evidence:**

The paper provides convincing empirical evidence that their proposed method works. Most smaller claims along the way are well supported, except the quick but interesting discussion about "planning delusion". I would like to see some more robust discussion / specific experimentation backing up this point (more that provided in the Appendix), which feels quick weak.

**Essential References Not Discussed:**

I believe that there are a couple of missing refereces, but scholarship is on the whole good.
1. The classic meta-learning optimiser paper: https://arxiv.org/abs/1611.03824
2. Work using a very similar NN arcitecture + meta-training over GP samples, but with the goal of accelerating model fitting https://arxiv.org/abs/2106.08185
3. Again, similar meta-training models attempting to mimic GPs through a similar NN architecture: https://arxiv.org/abs/1910.13556
4. A non-myopic meta-learning approach to BO: https://arxiv.org/abs/2302.11533

**Experimental Designs Or Analyses:**

Aside from Rollout VR, why are there not more non-myopic algorithms compared agaisnt? This requires justification in my eyes.

**Methods And Evaluation Criteria:**

The banchmark datasets and experimental protocols seem valid + experiemntal details seem clear.

**Other Comments Or Suggestions:**

Have you though about building your RL environment using the SAASBO priors to sample your GP trajectories? This baseline seems to do well and it would be interesting if your method could learn to mimic some aspects of the behaviour + make it nonmyopic?

**Other Strengths And Weaknesses:**

The paper seems novel and makes an important contribution. The paper is also very well written and was a pleaseure to read.

**Questions For Authors:**

I would be tempted to increase my score if the following two aspects could be addressed:
1) I want to see performance if you just train the RL agent on the GP prior (i.e. no step-sepcfici training) and add this as a baseline. This would really help provide context for how well your step-based episodic environments work
2) improved exploration of your hypothesis of planning delusion

**Relation To Broader Scientific Literature:**

This paper sits within a popular trend of armortising certain prohibitively expensive operations with GP models, particulary in the context of optimisation. Scholarship is mainly good, but a couple of references are missing (see below).

I do feel like this is a substantial step forward, providing a general framework for deploying complicated BO. I imagine that this work could spin out into lots of interesting future work, applying this framework to other difficult optimisation problems.

**Theoretical Claims:**

N/A

---

> ### Author Rebuttal · Authors · 2025-04-01
>
> We sincerely thank the reviewer for their thoughtful comments and positive feedback on our work. We appreciate the time taken to understand our contribution and the valuable suggestions to strengthen the paper.
>
> ### **Regarding the "planning delusion" concept:**
>
> We will conduct further experimentation on this concept as suggested. The text below is repeated from our response to Reviewer euH5:
>
> Regarding 'planning delusion,' or the effect of over-reliance on GP with high uncertainty, we agree this is an important component that warrants further exploration. We believe planning delusion occurs because EARL-BO repeatedly samples from the GP model, adds them to the dataset, updates the GP, and so on. This iterative process leads to increasingly less reliable GP models and RL environments further into the lookahead horizon. This explains the sub-optimal performance with excessively long lookahead horizons.
>
> We will include additional experiments testing 1 (i.e., standard myopic), 3, 5, and 7-step lookahead EARL-BO variants on the benchmark functions. If our hypothesis is correct, we expect the smaller lookahead horizons to perform better at earlier iterations, when the initial GP is more uncertain, but longer lookahead horizons to dominate at later iterations, when the initial GP is more representative of the black-box function. Our preliminary findings on the 8D Ackley function indeed follow this pattern: in earlier iterations, the 3-step lookahead method performs best; however, as optimization progresses, the 5-step lookahead method shows superior performance. The 7-step lookahead method performs worse consistently, suggesting the GP environment never reaches an accuracy amenable to such a long lookahead horizon. We will include and expand on these ablation study results in a revised version.
>
> Incorporating uncertainty-aware lookahead mechanisms is an excellent idea for future expansion. We agree that real and virtual data should be treated with different levels of trust. Uncertainty quantification would help mitigate planning delusion by accounting for the reliability of predictions, e.g., with a robust BO approach. Additionally, an adaptive horizon approach (using longer lookahead when uncertainty is low) could be a promising direction for future research.
>
> ### **On the comparison with other non-myopic algorithms:**
>
> We aimed to compare against N-step lookahead BO methods (N>2) that are general and can be applied to non-trivial black-box optimization problems (e.g., handling ~10 dimensions). The Rollout method with variance reduction represents the most recent approach meeting these criteria. Many other non-myopic methods face scalability challenges in higher dimensional spaces, making fair comparisons difficult (though we appreciate it if the reviewer can point us to suitable methods). Nevertheless, we appreciate this feedback and will clarify this justification in the revised manuscript.
>
> ### **Regarding the RL agent trained on GP prior only:**
>
> We are admittedly uncertain what the reviewer means by "no step-specific training." We took this to mean RL-based BO without lookahead steps. Our ablation study for planning delusion will include experiments with exactly 1-step RL-based BO. We expect these results to show that performance is comparable to EI-based BO, since both methods effectively optimize for immediate improvement (RL effectively serves as the "optimizer" for the EI acquisition function). Adding this ablation study will support that the multi-step lookahead approach is the main contributor to EARL-BO's enhanced optimization efficiency rather than the RL component. We will include this helpful analysis in a revised version.
>
> ### **On using SAASBO priors:**
>
> Thank you for this insightful suggestion. We envision EARL-BO as a flexible and modular framework for RL-based BO, which can be fine-tuned to particular problem settings. It can conduct off-policy learning with any state-of-the-art BO method, including SAASBO, and suggest points that will have optimal multi-step lookahead effects. Moreover, other aspects could also be switched, e.g., the choice of RL algorithm (see response to Reviewer eu5H). These directions indeed represent exciting avenues for future work, and we will highlight this in the revised manuscript.
>
> We thank the reviewer again for their constructive feedback and are pleased that they found our paper well-written and a "substantial step forward" in the field that can help with many difficult optimization problems.

---

> > ### Comment · Reviewer_JCQV · 2025-04-02
> >
> > Please can you discuss how your work relates to the additional references that I provided?
> >
> > I am now increasingly worried about the computational cost issues raised by the other reviwers. This is something I didn't quite notice when I read the paper. These high costs do make this algorithm very difficult to use in practice and somewhat undermine my postive comment "I do feel like this is a substantial step forward, providing a general framework for deploying complicated BO. I imagine that this work could spin out into lots of interesting future work, applying this framework to other difficult optimisation problems."
> >
> > Do the authors have anything to suggest about in what settings people would likely use this algorithm?

---

> > > ### Author Response · Authors · 2025-04-08
> > >
> > > Thank you for acknowledging our scholarship efforts and pointing us to the omitted references. We will indeed expand our discussion of the broader context using these in a revised version. In particular:
> > >
> > > - [1] This paper is a seminal effort in meta-learning for black-box optimization and will be discussed when introducing meta-learning for Bayesian optimization (the authors nicely contrast their method with BO).
> > > - [2-3] These works use similar attention-based architectures to encode datasets, although for other applications as noted. These share commonalities to our method in their encoder, which takes in a full dataset with the aim of learning and encoding of statistics of a full stochastic process (cf. our training is purely task driven). We will discuss these references when introducing the learned representation of the dataset.
> > > - [4] This recent work discusses how meta-learning can be applied to the slightly different setting of movement-cost constrained BO, which the authors nicely compare to non-myopic BO (Section 4). We will include this line of research in our discussion of meta-learning for BO.
> > >
> > > ### [Applicability of EARL-BO]
> > > In response to the reviewers' concerns, we will highlight the relative efficiency of the EARL-BO framework using the 30-dimensional Ackley function. A preliminary version of this experiment (two random starts instead of ten for now) can be found at https://anonymous.4open.science/r/icml_2025_review-B587. EARL-BO significantly outperforms all comparison methods, again demonstrating its scalability to challenging optimization problems in practice.
> > >
> > > The computational requirements for this added experiment emphasize our scalability results: the rollout-based method would require even more Monte Carlo iterations to handle this search space, yet using even 1000 iterations already times out in the 3600s budget. We believe non-myopic methods are the best computational comparison point (see discussions with other reviewers). On the other hand, EARL-BO runs in approximately 1600s, a moderate increase in time compared to the much less complicated setting of optimization over the 8-D Ackley function. Specifically, compared to 8-D 3-step lookahead EARL-BO, BO iterations in the 30-D problem require approximately 100% more CPU time, while myopic methods, such as EI and TuRBO require 533% and 215% more CPU time, respectively, to make decisions in 30-D compared to the 8-D optimization problem. We believe these results strengthen our contribution and will include a more comprehensive analysis in a revised version.
> > >
> > > Despite strongly outperforming rollout-based methods with less time, we acknowledge the reviewers’ concerns regarding computational efficiency compared to the cheaper (myopic) methods. We do acknowledge this limitation of EARL-BO (and in fact non-myopic methods in general), but note that **EARL-BO, and BO in general, is specifically designed for optimization problems where function evaluations are expensive,** such as hyperparameter tuning for ML models [1], engineering design/simulations [2], and chemical laboratory experiments [3]. In these settings, a single function evaluation can require hours or even days, not to mention monetary costs, while the overhead of BO remains relatively small (difficulty of these problems is also mentioned the Reviewer). Thus, even as EARL-BO introduces additional computational cost relative to *myopic* baseline BO approaches, its significant improvement in performance can justify this expense. Furthermore, the improved sample efficiency of our method means that fewer function evaluations are required to reach an optimal or near-optimal solution. This results in a (significant) net reduction in total cost when considering the entire optimization process. Future works can exploit the modularity of EARL-BO for faster settings, e.g., by warm-starting RL, alternative RL methods/tunings, using the actor for multiple BO iterations, etc.
> > >
> > > **References:**
> > >
> > > [1] Snoek, J., Larochelle, H., & Adams, R. P. (2012). Practical Bayesian optimization of machine learning algorithms. NeurIPS 2012.
> > >
> > > [2] Jones, D. R., Schonlau, M., & Welch, W. J. (1998). Efficient global optimization of expensive black-box functions. Journal of Global Optimization, 13, 455-492.
> > >
> > > [3] Shields, B. J., Stevens, J., Li, J., Parasram, M., Damani, F., Alvarado, J. I. M., Janey, J. M., Adams, R. P., & Doyle, A. G. (2021). Bayesian reaction optimization as a tool for chemical synthesis. Nature, 590(7844), 89-96.

---

### Official Review · Reviewer_euH5 · 2025-03-14

**Overall Recommendation:** 3

**Summary:**

The paper introduces EARL-BO, a novel reinforcement learning (RL)-based framework for multi-step lookahead Bayesian Optimization (BO) in high-dimensional black-box optimization problems.

**Claims And Evidence:**

The paper makes several key claims:
- EARL-BO improves multi-step lookahead BO performance in high-dimensional settings.
  - Evidence: Experiments on synthetic and real-world HPO tasks show that EARL-BO outperforms single-step BO methods and achieves comparable or better results than rollout-based BO in low dimensions.
- EARL-BO is scalable to high-dimensional problems.
  - Evidence: The use of an Attention-DeepSets encoder and RL-based optimization allows it to handle up to 19D problems, unlike traditional rollout-based methods which are limited to low-dimensional cases.
- EARL-BO mitigates myopic behavior in BO.
  - Evidence: Experimental results demonstrate that multi-step lookahead strategies improve long-term optimization results, particularly in complex and higher-dimensional search spaces.
- The GP-based virtual environment enables efficient learning.
  - Evidence: The model-based RL approach significantly improves sample efficiency, reducing reliance on expensive function evaluations.

While these claims are largely supported, some concerns include:
- Planning Delusion: The paper acknowledges that EARL-BO can suffer from poor decision-making when the GP posterior has high uncertainty, especially in high-dimensional settings with sparse data.
- Computational Cost: Although EARL-BO is scalable, the training overhead of RL-based BO is not extensively compared to other methods.

**Essential References Not Discussed:**

None

**Experimental Designs Or Analyses:**

The experiments are well-structured and include:
- Multiple replications (10 runs per experiment).
- Ablation studies (e.g., lookahead horizon effects, learning rate sensitivity).
- Comparisons against strong baselines.

However, there are some limitations:
- Lack of runtime comparisons: How does EARL-BO’s computational cost compare to TuRBO or SAASBO?
- Planning Delusion Analysis: The effect of high uncertainty in GP modeling on EARL-BO's decision-making could be explored further.

**Methods And Evaluation Criteria:**

The proposed methods are well-suited for multi-step lookahead BO and high-dimensional black-box optimization. The evaluation criteria include:
- Synthetic benchmark functions (Ackley, Levy, Rosenbrock, Sum Squares) tested in 2D, 5D, and 8D.
- Real-world HPO tasks (HPO-B dataset) with search spaces 6D, 8D, and 19D.
- Comparison against state-of-the-art BO methods including EI, TuRBO, SAASBO, and rollout-based methods.

The benchmarks and criteria are appropriate, but additional details on computational efficiency and runtime analysis would strengthen the evaluation.

**Other Comments Or Suggestions:**

- Provide a runtime comparison with existing BO methods.
- Analyze how EARL-BO scales computationally with increasing dimensions.
- Explore potential improvements to mitigate planning delusion (e.g., uncertainty-aware lookahead adjustments).

**Other Strengths And Weaknesses:**

Strengths:
- Innovative integration of RL with BO for multi-step lookahead.
- Scalability to high-dimensional problems.
- Strong empirical validation across synthetic and real-world tasks.
- Comprehensive ablation studies.

Weaknesses:
- Computational cost is not analyzed in depth.
- Limited discussion on theoretical RL convergence properties.

**Questions For Authors:**

- How does EARL-BO’s training cost compare to TuRBO or SAASBO?
  - A direct runtime comparison would clarify its practical applicability.
- Would an uncertainty-aware lookahead mechanism help mitigate planning delusion?
  - Given that long-horizon lookahead can lead to compounding GP errors, would incorporating uncertainty quantification help?
- How does EARL-BO perform when initialized with a poorly trained GP?
  - If the GP model is unreliable in early iterations, does EARL-BO suffer significant performance degradation?
- Could alternative RL methods (e.g., model-free RL) be competitive?
  - PPO is used, but would Q-learning or offline RL approaches provide benefits?

**Relation To Broader Scientific Literature:**

The paper is well-situated within the Bayesian optimization and reinforcement learning literature. The connections are well-documented, but a deeper discussion on the trade-offs between model-based vs. model-free RL approaches in BO would be valuable.

**Theoretical Claims:**

The paper formulates BO as a finite-horizon Markov Decision Process (MDP) and proposes an RL-based solution. The theoretical framework is sound, leveraging:
- Dynamic programming to justify multi-step optimization.
- Attention-DeepSets encoders for permutation invariance.
- PPO-based RL policy optimization.

The theoretical justification appears correct, though more formal convergence guarantees for the RL policy would be beneficial.

---

> ### Author Rebuttal · Authors · 2025-04-01
>
> We appreciate your thoughtful review and valuable suggestions for improving our work. Your feedback is extremely helpful, and we are grateful for your careful consideration of our work.
>
> ### **Planning Delusion and Uncertainty-Aware Mechanisms**
>
> Regarding 'planning delusion,' or the effect of over-reliance on GP with high uncertainty, we agree this is an important component that warrants further exploration. We believe planning delusion occurs because EARL-BO repeatedly samples from the GP model, adds them to the dataset, updates the GP, and so on. This iterative process leads to increasingly less reliable GP models and RL environments further into the lookahead horizon. This explains the sub-optimal performance with excessively long lookahead horizons.
>
> We will include additional experiments testing 1 (i.e., standard myopic), 3, 5, and 7-step lookahead EARL-BO variants on the benchmark functions. If our hypothesis is correct, we expect the smaller lookahead horizons to perform better at earlier iterations, when the initial GP is more uncertain, but longer lookahead horizons to dominate at later iterations, when the initial GP is more representative of the black-box function. Our preliminary findings on the 8D Ackley function indeed follow this pattern: in earlier iterations, the 3-step lookahead method performs best; however, as optimization progresses, the 5-step lookahead method shows superior performance. The 7-step lookahead method performs worse consistently, suggesting the GP environment never reaches an accuracy amenable to such a long lookahead horizon. We will include and expand on these ablation study results in a revised version.
>
> Incorporating uncertainty-aware lookahead mechanisms is an excellent idea for future expansion. We agree that real and virtual data should be treated with different levels of trust. Uncertainty quantification would help mitigate planning delusion by accounting for the reliability of predictions, e.g., with a robust BO approach. Additionally, an adaptive horizon approach (using longer lookahead when uncertainty is low) could be a promising direction for future research.
>
> ### **Computational Cost Analysis**
>
> We agree that runtime comparisons are also a crucial consideration that warrant detailed investigation. We provide computational cost results for the 8D Ackley function below:
>
> | Method | Average Runtime (s) | Std Dev (s) | Notes |
> | --- | --- | --- | --- |
> | EI | 0.28 | 0.05 | - |
> | TuRBO | 0.27 | 0.20 | - |
> | SAASBO | 168.6 | 27.6 | - |
> | Rollout_EI (3-step) | >3600 | - | 1000 MC iterations |
> | EARL-BO (3-step) | 840 | 147 | 4000 episodes |
> | EARL-BO (5-step) | 1075 | 134 | 4000 episodes |
>
> While multi-step lookahead methods inevitably require more computation time, we note that Bayesian optimization is typically applied to problems where function evaluations are expensive (e.g., materials discovery, hyperparameter tuning of large models, etc.). Nevertheless, it is important to highlight these performance tradeoffs, and we will include a comprehensive table of CPU times in the final version.
>
> ### **Performance with Poorly Trained GPs**
>
> We assume that poorly trained GPs affect all BO methods. However, we hypothesize the effects are more pronounced for multi-step lookahead methods, because looking multiple steps ahead via an untrustworthy environment may cause sub-optimal performance. However, as shown in our appendix experiment with the 19D optimization problem, using only 5 initial points (creating a challenging sparse data scenario, which can lead to poor choice of hyperparameters), EARL-BO initially struggles but eventually finds the global optimum faster than traditional methods such as EI, PI, UCB, and TuRBO. This demonstrates the robustness of our approach even in these challenging scenarios.
>
> ### **Alternative RL Methods**
>
> The question about alternative RL methods is rather insightful. In fact, off-policy methods might offer sample efficiency advantages in certain scenarios. As our intention is to present EARL-BO as a novel framework, rather than to tune to a particular application, we select PPO due to its well-documented stability, robustness to hyperparameters, and strong empirical performance in continuous action space tasks similar to our application. We will emphasize that the EARL-BO framework is agnostic to the choice of RL method, and the optimal design of an RL method represents an interesting direction for future research that could yield further improvements.
>
> We appreciate your constructive feedback and will incorporate your suggestions to strengthen the paper.

---

> > ### Comment · Reviewer_euH5 · 2025-04-03
> >
> > Computational efficiency remains a major concern for me. The time cost of your EARL-BO is indeed quite high. Please provide a detailed explanation of what *Average Runtime (s)* specifically represents—is it the time for a single estimation, a single iteration, or something else? Comparing the computation time of iterations and episodes seems unfair. Please ensure that all baselines are evaluated under a unified standard for computational efficiency.

---

> > > ### Author Response · Authors · 2025-04-08
> > >
> > > Thank you for the continued engagement and interest in our work. We apologize for the unclear presentation of our preliminary results. We do believe “all baselines are evaluated under a unified standard for computational efficiency,” as explained in the following.
> > >
> > > ### [Analysis of Runtimes]
> > > The table of preliminary computational cost analysis reports the average runtime for *each Bayesian optimization iteration* following the respective frameworks. The notes column was intended to reflect some important hyperparameter choices, rather than to suggest we compare iterations against episodes, estimations, etc. We will include more comprehensive results and analyses in a revised version, including:
> > >
> > > 1. EI, TuRBO, and SAASBO are much faster, but are “myopic” in the sense they do not aim to solve BO as an SDP;
> > > 2. Rollout_EI is run with 1000 Monte Carlo iterations, which is still a small number, given we had to use similar numbers for the 2- and 4-D problems presented in the paper. Monte Carlo methods require a number of samples scaling with the search space and scenario tree (here based on dimensionality and lookahead horizon) to achieve reasonable accuracy, which becomes impractical as dimensionality increases;
> > > 3. Modern RL methods employ function approximation techniques such as actor-critic networks, which can learn useful policies from far fewer samples by leveraging generalization, rather than exhaustive sampling (see below on scalability). We found PPO to perform well with only 4000 episodes, but again note that our EARL-BO framework can integrate alternative/better RL algorithms seamlessly, e.g., by warm-starting RL, alternative RL methods/tunings, using the actor for multiple BO iterations, etc.
> > >
> > > ### [Scalability of EARL-BO]
> > > To further highlight the relative efficiency of the EARL-BO framework, we will include optimization results on the 30-dimensional Ackley function. A preliminary version of this experiment (two random starts instead of ten for now) can be found at https://anonymous.4open.science/r/icml_2025_review-B587. EARL-BO significantly outperforms all comparison methods, again demonstrating its scalability to challenging optimization problems in practice.
> > >
> > > The computational requirements for this added experiment emphasize our scalability results: the rollout-based method would require even more Monte Carlo iterations to handle this search space, yet using even 1000 iterations already times out in the 3600s budget. On the other hand, EARL-BO runs in approximately 1600s, a moderate increase compared to the much less complicated setting of optimization over the 8-D Ackley function. Specifically, compared to 8-D 3-step lookahead EARL-BO, BO iterations in the 30-D problem require approximately 100% more CPU time, while myopic methods, such as EI and TuRBO require 533% and 215% more CPU time, respectively, to make decisions in 30-D compared to the 8-D optimization problem. We believe these results strengthen our contribution and will include a more comprehensive analysis in a revised version.
> > >
> > > Despite strongly outperforming rollout-based methods with less time, we acknowledge the reviewers’ concerns regarding computational efficiency compared to the cheaper (myopic) methods. We do acknowledge this limitation of EARL-BO (and in fact non-myopic methods in general), but note that BO is specifically designed for optimization problems where function evaluations are expensive, such as hyperparameter tuning for ML models [1], engineering design/simulations [2], and chemical laboratory experiments [3]. In these settings, a single function evaluation can require hours or even days, not to mention monetary costs, while the overhead of BO remains relatively small (difficulty of these problems is also mentioned by Reviewer JCQV). Thus, even as EARL-BO introduces additional computational cost relative to *myopic* baseline BO approaches, its significant improvement in performance can justify this expense. Furthermore, the improved sample efficiency of our method means that fewer function evaluations are required to reach an optimal or near-optimal solution. This results in a (significant) net reduction in total cost when considering the entire optimization process. Future works can exploit the modularity of EARL-BO for faster settings, as noted above.
> > >
> > > **References:**
> > >
> > > [1] Snoek, J., Larochelle, H., & Adams, R. P. (2012). Practical Bayesian optimization of machine learning algorithms. NeurIPS 2012.
> > >
> > > [2] Jones, D. R., Schonlau, M., & Welch, W. J. (1998). Efficient global optimization of expensive black-box functions. Journal of Global Optimization, 13, 455-492.
> > >
> > > [3] Shields, B. J., Stevens, J., Li, J., Parasram, M., Damani, F., Alvarado, J. I. M., Janey, J. M., Adams, R. P., & Doyle, A. G. (2021). Bayesian reaction optimization as a tool for chemical synthesis. Nature, 590(7844), 89-96.

---

### Decision · Program_Chairs · 2025-05-01

**Decision:**

Accept (poster)

**Comment:**

This manuscript proposes a new methodology (called EARL-BO) for nonmyopic Bayesian optimization grounded in reinforcement learning. The authors study the performance of this method in an empirical study involving both synthetic and real objective functions, where it was competitive with alternative approaches.

The initial response to this work from the reviewers was somewhat mixed, although leaning positive. The reviewers generally found the material to be of interest to the ICML audience and for the proposed methodology to be well-motivated and clear. The reviewers did have some questions and concerns, which were mostly addressed satisfactorily by the author responses.

This paper generated significant discussion during the discussion phase. The primary issue raised, which was seen as a blocker by one reviewer, was the computational cost of the prosed methodology. However, nonmyopia by its very nature comes with increased computational cost, and EARL-BO does appear to be competitive in performance and runtime to other nonmyopic alternatives. Thus it likely lies somewhere on the Pareto frontier of runtime vs performance. This, combined with the potentially interesting ideas put forward in the manuscript, is sufficient to warrant publication in my opinion.